# Violent Content in Online Pornography Is Associated with Sexual Health of Women and Men

**DOI:** 10.3390/bs15121634

**Published:** 2025-11-27

**Authors:** Belén Sanz-Barbero, Vanesa Pérez-Martínez, Ana Rico, Laura Otero-García, Marta Fernández-López, Ariadna Cerdán-Torregrosa, Carmen Vives-Cases

**Affiliations:** 1National School of Public Health, Institute of Health “Carlos III”, 28029 Madrid, Spain; bsanz@isciii.es (B.S.-B.); arico@isciii.es (A.R.); laura.otero@isciii.es (L.O.-G.); 2CIBER of Epidemiology and Public Health (CIBERESP), 28029 Madrid, Spain; carmen.vives@ua.es; 3Public Health Research Group, Department of Community Nursing, Preventive Medicine and Public Health, History of Science, University of Alicante, 03690 Alicante, Spain; mfl32@gcloud.ua.es (M.F.-L.); ariadna.cerdan@ua.es (A.C.-T.); 4Department of Sociology II, University of Alicante, 03690 Alicante, Spain

**Keywords:** online violent pornography, sexual health, empathy, sexual assertiveness, acceptance of violence against women

## Abstract

Sexual health includes psychosocial and physical competencies related to sexual well-being. It is unknown whether violent pornography is associated with sexual health. The objective of this study is to analyze whether different types of pornography are associated with sexual health in young adults. A cross-sectional study was conducted using an online survey with 3607 people aged 18–35 living in Spain. Association of pornography with the dependent variables was estimated with generalized linear and logistic models. Compared to exposure to pornography with no explicit violence, men exposed to pornography with physical violence scored lower on empathy (coef: −0.054) and sexual assertiveness (coef: −0.034) and higher on acceptance of violence against women (coef: 0.122). Women who watch physical violence pornography scored higher on acceptance of violence against women (coef: 0.076). In men, watching pornography with physical violence was associated with poorer sexual function (OR = 1.9). In women, pornography with physical violence was associated with difficulty having an orgasm in a relationship (OR = 1.7). Violent pornography could have wide negative effects on men sexual health. In both sexes, violent pornography was associated with greater acceptance of violence against women. Inequalities were observed between women and men in the enjoyment of sexual relations, to the detriment of women.

## 1. Introduction

In 2006, the World Health Organization ([81]) conceptualized sexual health as a condition of physical, emotional, mental, and social well-being related to sexuality, which goes beyond the mere absence of disease, dysfunction, or infirmity. Furthermore, it stated that sexual health entails adopting a positive and respectful attitude toward sexual relationships, as well as the capacity to engage in safe and satisfying sexual experiences free from coercion, discrimination, and violence. Therefore, sexual health includes social, psychological and physical skills related to sexuality, such as empathy, sexual assertiveness, sexual satisfaction, sexual response and the absence of sexually transmitted infections (STIs) ([81]; [78]). Some of these skills are essential to develop sexual health since they are related to both emotional aspects and sexual response ([28]; [45]; [68]).

Sexual self-esteem is positively correlated with sexual function ([73]), sexual satisfaction and sexual assertiveness ([53]) in men and women. Low sexual self-esteem is a vulnerability factor for sexual victimization in women ([47]; [70]; [76], [77]), as well as for aggressive sexual behaviour in both sexes ([70]). In addition to being a fundamental part of sexual function and sexual response ([53]; [66]), sexual assertiveness has been identified as a protective factor against sexual victimization and coercion ([49]), as well as against risky sexual behaviours ([35]; [72]). Empathy is also considered key in terms of psychosocial competencies. Literature shows an association between greater empathy and greater sexual satisfaction in both men and women ([28]). Higher levels of empathy are also associated with a lower likelihood of aggressive sexual behaviour ([39]).

Due to the role of social and psychological skills in sexual development and interpersonal relationships, Spanish legislation includes affective-sexual education transversally during compulsory schooling years (Spanish Organic Law 3/2020, [9]) as a mandatory part of the education system. Despite this and given the increase in crimes against sexual freedom committed by minors in Spain in recent years ([57]), as well as the high consumption of pornography from a very young age ([5]; [69]), affective-sexual education is in high demand in society.

Although scientific evidence is heterogeneous regarding the possible association of pornography consumption with victimization and aggressive sexual behaviour ([22]; [83]), preliminary evidence shows that pornography consumption is associated with social skills that mediate the development of sexuality. Specifically, problematic pornography consumption in young men has been associated with lower levels of empathy ([25]; [46]). Recent studies show that women who consume pornography have lower sexual self-esteem ([24]) and this is associated with the development of pornography addiction in men ([71]). Despite the aforementioned evidence, the association of pornography with sexual satisfaction and sexual function is heterogeneous. Studies with women showed that using pornography as a couple is associated with greater sexual function and less sexual distress ([11]). In men, perceived addiction to pornography has been negatively associated with sexual function ([80]). A recent meta-analysis identified a negative correlation between pornography use and sexual satisfaction in women, yet this was not found in men ([1]).

Although there has been an increase in recent years in studies that analyze the association of pornography consumption with acceptance of violence against women (Accept-VAW) ([14]), evidence on the possible association of pornography consumption with sexual health is still scarce. Moreover, the Antecedents-Context-Effects Model (ACE) ([15]) suggests that the effect of pornography consumption on sexual well-being largely depends on the context, which includes the pornographic content that is viewed. In accordance with this model, there is evidence that the association of pornography and aggressive behaviour ([65]; [83]) and sexual victimization is more consistently related to watching violent pornography ([64]; [65]; [83]). However, one aspect that remains unclear is how watching different types of violent pornography is connected to sexual health.

Based on the above, this study hypothesizes that the association between watching pornography and sexual health varies depending on its violent content. More specifically, it is possible that people who watch violent pornography have worse sexual health compared with people who watch pornography with no explicit violence. This effect could be different in women than in men, depending on the different dimensions and competencies of sexual health studied. Specifically, and based on the evidence presented above, exposure to violent pornography may be associated with: (a) lower sexual assertiveness in women and men; (b) lower empathy in men; (c) greater Accept-VAW in both sexes and; (d) a higher prevalence of sexually transmitted infections in both sexes. The direction of the possible association of exposure to violent pornography with sexual function is unknown.

Therefore, the objective of this study is to analyze the association of exposure to different types of pornography -with no explicit violence (PNV), degrading/humiliating practices (PDH) and with physical violence (PPhV)—with empathy, sexual assertiveness, sexual satisfaction, Accept-VAW, sexual response and STIs in young adult women and men.

## 2. Materials and Methods

### 2.1. Sample and Participants

Cross-sectional study of the survey “Pornography use, sexual health and sexual violence” conducted among 3607 women and men aged between 18 and 35, living in Spain. In order to calculate the sample size, we took a frequency of pornography use in the last 12 months of 75% in the population aged between 18 and 35 as a reference ([69]). We set the lower limit at 18 years because that is the age of majority in Spain, which facilitates the collection of data. Furthermore, legal access to pornographic websites is also permitted at age 18. We set the upper limit at 35 years old because, given that access to online pornography became widespread around the year 2000 ([11]), individuals between 18 and 35 years old could have had easy access to online pornography from the beginning of their sexual development. For an expected OR equal to 1.5, a confidence level of 95% and a power of 80%, a minimum sample size of 3564 questionnaires was estimated, with a relative precision of 25.4%. The sample design, participant recruitment, and data collection were carried out by an external company specializing in social studies. The questionnaire was conducted online. The sample was designed based on a panel of volunteers including 19,000 people from the age group of this study. The panel, Netquest, complied with the ISO 26362:2009 quality standard for the management of research panels ([58]). A quota sample was designed considering the distribution of the Spanish population for this age group according to age, sex and region of residence. In order to meet age quotas, weighting coefficients were included for this variable. An email was sent to potential participants (*n* = 9125) asking them if they were interested in answering a questionnaire on social issues. In total, 5235 people answered that they were interested. However, 1221 were rejected since they did not meet the quotas when they answered. From the 4014 people who began answering the questionnaire, 3607 finished it after passing different quality control filters. Participation in the study was voluntary, and volunteers did not receive financial remuneration but were compensated with vouchers that could be exchanged for items listed in a catalog provided by the company collecting the data (home, computers, leisure, stationery, etc.). This study analyses a subsample of 1985 sexually active individuals who report having watched pornography in the past 12 months.

The study was conducted according to the guidelines of the Declaration of Helsinki and approved by the Ethics Committee of the Carlos III Health Institute (Exp: CEI PI 04_2023-v3) and all participants signed an informed consent beforehand.

### 2.2. Measurement Instrument

#### 2.2.1. Dependent Variables

Well-being, sexual function and STIs: The items analyzed originate from the instrument “Female sexual health and dysfunctions in primary care” in the sexual function domain (items 13, 14, 15, 16, 17 and 19). The instrument was designed and validated by [4] ([4]) in Spain in a sample of women over 18 years of age. Questionnaire validation showed good psychometric properties; therefore, it is a suitable questionnaire to approach women’s sexual health in the healthcare setting.

In order to be used with both women and men, the following ad hoc questions were included: (a) Do you have difficulty getting or maintaining an erection? (b) Do you experience premature ejaculation? and (c) Do you use sexual enhancers? In addition, the following ad hoc items were included for both sexes: (a) Do you have fun during or enjoy sexual activity? (b) Do you sometimes feel anxiety, fear or other discomfort during sexual activity? and (c) How would you rate yourself as a regular sexual partner on a scale of 1–10? All items were answered on a Likert scale from 1 to 6, with 1 being never/not at all and 6 always/a lot, depending on the item. For further analysis, the responses were dichotomized. Cronbach’s alpha was 0.79 for women and 0.78 for men. Both men and women were asked if they had been diagnosed with an STI in the past 12 months. If they answered affirmatively, they were asked to specify the infection.

Empathy: The items used are taken from the Brief Basic Empathy Scale (BES-B), specifically items 2 and 7. The scale was originally defined by [43] ([43]). The short version of the scale (BES-B) was developed by [61] ([61]) (BES-B) and uses nine items to assess two dimensions: affective empathy and cognitive empathy. Item 2 is included in cognitive empathy and item 7 in affective empathy. Participants were asked to indicate their level of agreement with the items on a scale from 1 (strongly disagree) to 5 (strongly agree). The BES-B survey was evaluated in a sample of adolescents over 12 years of age, demonstrating good psychometric properties. This study analyses the sum of items 2 and 7. Higher scores indicate greater empathy. Cronbach’s alpha for the items was 0.59.

Sexual assertiveness: The items used are part of the Hurlbert Index of Sexual Assertiveness (HISA), originally described by [40] ([40]), but adapted and validated by [67] ([67]) in women and men aged between 18 and 71. This index includes 19 items answered on a 5-point Likert-type scale. The scale includes two dimensions: (a) initiation, understood as the ability to initiate a sexual activity; and (b) lack of shyness/rejection, defined as the absence of shyness or ability to reject an unwanted sexual activity. Our study includes and analyses the sum of items 1, 9, 11 and 19, all of which are from the lack of shyness/rejection dimension. Higher scores indicate greater sexual assertiveness. Cronbach’s alpha for the items was 0.94.

Acceptance of violence against women (Accept-VAW): The items included in this study were used by the Centro de Investigaciones Sociológicas (CIS) in the study “Survey on sexual violence against women” ([17]), aimed at women and men over 18 years of age living in Spain, as well as in the study “Social perception of sexual violence” ([16]), aimed at a population of women and men over 16 years and living in Spain. The response categories are acceptable in some circumstances (3), unacceptable but should not be punished by law (2) and unacceptable and should always be punished by law (1). Higher scores indicate greater acceptability of VAW. Cronbach’s alpha for the items was 0.83.

#### 2.2.2. Independent Variable

The main independent variable was pornography consumption in the last 12 months. An ad hoc 12-item instrument was designed to collect information on pornography viewed online in the last 12 months (see Appendix A). Digital/online pornography was defined as “watching sexually explicit audiovisual material (photographs, videos, short films, films) on the Internet (computer, tablet, television or mobile phone)”. To reduce social desirability bias, the items were worded to avoid words such as rape or pain, among others. To facilitate self-reporting of violent practices, the items were rated on a 5-point Likert scale ([3]). The responses were dichotomized for the analysis. Each item contains 3 or 4 subitems that identify the sex of the people who interact with each pornographic practice. For analytical purposes, pornography consumption was recoded into a single variable with three mutually exclusive categories: (a) people who exclusively watch pornography with no explicit violence (PNV) (e.g., viewing consensual sexual relationships without the use of force); (b) people who watch pornography with depictions degrading/humiliating practices but without physical violence (PDH) (e.g., sexual relationships involving insults or threats; practices of domination and submission without causing physical pain); (c) people who watch pornography with depiction of physical violence practices (PPhV) (e.g., depictions of non-consensual sexual relationships involving the use of force; Bondage-Discipline-Domination-Submission involving painful practices, etc.). The description of the items can be found in Appendix A.

The instrument was designed by five researchers who are experts in gender-based violence, sexual violence and pornography use. Once the instrument was designed, it was qualitatively validated with the study population. First, a focus group was created where nine people (between 21 and 34 years of age), all of them with degrees related to social sciences or health sciences, completed the questionnaire and discussed each of the items in the instrument. The discussion included understanding the items, the appropriateness of the terms used in them, the time frame and the subsequent classification of the items into broader categories. Subsequently, 15 people aged between 18 and 35 completed the questionnaire and drafted an individual report on their understanding and acceptability of the items. Once that phase was finalized, the research team agreed on the final instrument.

Cronbach’s alpha of the instrument was 0.80 and 0.87 for the sample of women and men, respectively, indicating good internal reliability. The latent structure of the items assessing different types of pornography was analyzed through factor analysis, using principal component analysis with varimax rotation for extraction. The three responses to the pornography variable (PNV; PDH; PPhV) showed a unidimensional latent structure, therefore there was no component rotation. The Bartlett test of sphericity was <0.001 for each of the item groups in both women and men. The percentage of variance explained in women was PNV: 43%; PDH: 59%; PPhV: 59%. In men it was PNV: 41%; PDH: 66%; PPhV: 69%.

### 2.3. Covariates

Sociodemographic variables include age, as continuous variable; sex; completed university studies, categorized in yes/no; country of birth categorized in Spain/outside Spain.

Sexual orientation was defined based on the sex of the participant and the sex of the partners with whom they had sexual relations. The response categories were heterosexual/homosexual/bisexual. The relationship with sexual partners in the past 12 months included the following response categories: only stable partners, only casual partners, both stable and casual partners. Frequency of pornography use included the following categories: once a week or more, 2–3 times a month, once a month or less.

Finally, we adjusted by social desirability using the short version (18 items) of the Marlowe-Crowne Social Desirability Scale, which was validated in Spain by [32] ([32]).

### 2.4. Statistical Analysis

Firstly, a description of the sample is provided, stratified by sex and according to sociodemographic characteristics, sexual orientation and pornography consumption. The distribution of the dependent variables is described by the type of pornography consumed. The mean and standard deviation are described for empathy, sexual assertiveness and Accept-VAW. The effect size is calculated using Eta-squared values obtained through ANOVA tests. Eta-squared values of 0.01 are considered small, 0.06 medium, and 0.14 large ([18]). In the variables of well-being, sexual function and STIs, the frequency of the different items is described. The association of pornography usage with the dependent variables was conducted using generalized linear models in the continuous dependent variables (empathy, sexual assertiveness, Accept-VAW) and through logistic regressions in the categorical dependent variables (well-being, sexual function and STIs). All regression models were stratified by sex and adjusted according to level of education, sexual orientation, country of birth, type of sexual partner in the past 12 months, frequency of pornography usage in the past 12 months and social desirability.

All the analyses were performed using SPSS 29.0.2.0 ([41]) and Stata 18.0 ([74]).

## 3. Results

### 3.1. Sample Description

The sample analyzed includes women (*n* = 736) and men (*n* = 1249) aged 18–35 years, living in Spain, sexually active and reporting pornography consumption in the last 12 months. Table 1 describes the sociodemographic characteristics of the sample, as well as the type and frequency of pornography consumption in the last 12 months. In both sexes, the average age of the population was 29 years. 6.8% of respondents were born abroad, with this percentage higher among women (11.3%) than among men (3.2%). In addition, 11.6% of men and 2.4% of women have exclusively had same-sex partners, while 10.1% of men and 16% of women report partners of both sexes. Watching PPhV is more frequent among men (63.7%) than among women (50.7%). The frequency of pornography consumption is higher in men, with 62.1% reporting watching pornography once a week or more, compared to 14.9% of women.

### 3.2. Empathy, Sexual Assertiveness and Acceptance of VAW

Table 2 describes the mean values per variable of the sum of the different items conforming the variables empathy, sexual assertiveness and Accept-VAW, which were stratified by sex and the type of pornography viewed.

Men who watch PPhV have lower mean values of empathy (PPhV: 3.42 vs. PDH: 3.48 vs. PNV: 3.56) and assertiveness (PPhV: 4.24 vs. PDH: 4.44 vs. PNV: 4.43) and higher mean values of Accept-VAW (PPhV: 1.57 vs. PDH: 1.46 vs. PNV: 1.36) than those who watch other types of pornography.

Regarding women, those who watch PPhV show lower mean values of assertiveness than those who watch other types of pornography (PPhV: 4.23 vs. PDH: 4.43 vs. PNV: 4.41) and higher mean values of Accept-VAW (PPhV: 1.44 vs. PDH: 1.31 vs. PNV: 1.31). Effect sizes were small for all variables, for both women and men.

Table 3 shows the association of exposure to different types of pornography with the mean values of empathy, assertiveness, Accept-VAW and sexual self-esteem, taking the people who exclusively watch PNV as a reference. The results show that men who watch PPhV scored significantly lower on empathy measures [coef (95% CI): −0.054 (−0.015; −0.093)], as well as lower sexual assertiveness [coef (95% CI): −0.034 (−0.008; −0.061)]. Also in men, those who watch PDH [coef (95% CI): 0.059 (0.116; 0.002)], as well as those who watch PPhV [coef (95% CI): 0.122 (0.170; 0.073)] score significantly higher in acceptance of violence. No association was identified between the type of pornography consumed and sexual self-esteem or the esteem towards a sexual partner.

Regarding women, taking those who exclusively watch PNV as a reference, those who watch PPhV score significantly higher in Accept-VAW [coef (95% CI): 0.076 (0.029; 0.122)].

### 3.3. Sexual Health: Well-Being, Sexual Function and STIs

Table 4 presents sexual health outcomes in men and women according to type of pornography consumed. In men, taking those who watch PNV as a reference, those who watch PPhV report discomfort during sexual activity more frequently (quite a lot/a lot/very much: 14.1%); anxiety, fear or other discomfort (quite a lot/a lot/very much: 13.7%); difficulties in having an orgasm when they are with their partner, but not when they are alone (often, almost always, always: 10.6%), as well as greater difficulties in getting or maintaining an erection (often, almost always, always: 14.7%). Men who watch PPhV most frequently use sexual function enhancers (9.7%), followed by men who watch PDH (6.8%). Furthermore, 5.7% of men who watch PPhV have been diagnosed with an STI in the last year, compared to 1.1% of men who watch PNV. The most frequently mentioned STI diagnoses by men were gonorrhea (*n* = 17) and human papillomavirus (*n* = 12). Men who watch PPhV reported 76.5% of gonorrhea cases and 91.7% of papillomavirus cases (see Appendix A).

As for women (Table 4), those who watch PPhV report greater difficulties in having an orgasm when they are with their partner, but not when they are alone (10.6%). Additionally, 7.1% of women have been diagnosed with an STI in the last year, and this percentage increases to 8.3% among those who watch PPhV. The most frequently reported STI diagnoses among women were human papillomavirus (*n* = 24) and chlamydia (*n* = 9), reported in 54.2% and 77.8% of cases, respectively, by women who watch PPhV (see Appendix A).

Table 5 shows an association of the use of the different types of pornography with well-being, sexual function and STIs. In men, using those who watch PNV as a reference group, watching PPhV is associated with a greater probability of enjoying and having fun during sexual practices (OR = 2.4, 95% CI [1.1–5.4]), feeling discomfort during sexual activity (OR = 2.1, 95% CI [0.9–4.3]), and having difficulty in getting or maintaining an erection (OR = 1.9, 95% CI [1.0–3.4]). Watching PDH (OR = 4.1, 95% CI [1.1–14.8]) or PPhV (OR = 4.9, 95% CI [1.5–16.20]) is associated with a greater probability of using sexual function enhancers. On the other hand, taking women who watch PNV as a reference, watching PPhV was associated with a greater probability of reporting difficulties in having an orgasm in a relationship, but not alone (OR = 1.7, 95% CI [1.0–2.8]).

## 4. Discussion

### 4.1. Main Findings

There are differences in the pornography content used by women and men. A higher percentage of men watch pornography with physical violence, and a higher percentage of women watch degrading or humiliating pornography and pornography with no explicit violence. More than 50% of young adult women and men who have consumed pornography in the past year watch pornography with physical violence, although the frequency of pornography use per week is four times higher among men than among women.

This study identifies a negative correlation between watching violent pornography and certain well-being-related competencies in the sexual health of men and women. Men who watch pornography with physical violence scored significantly lower in empathy, in sexual assertiveness and higher in acceptance of violence against women than those who watched pornography with no explicit violence. Also, men who watch degrading or humiliating pornography scored higher in acceptance of violence against women. Regarding women, those who watch pornography with physical violence scored higher in acceptance of violence against women.

Furthermore, inequalities were observed between women and men regarding the associations of violent pornography consumption with other sexual health indicators. In men, considering those who watch pornography with no explicit violence as a reference, a positive association was identified between watching pornography with physical violence and the likelihood of enjoying and having fun during sexual encounters. A positive association was also identified with the likelihood of reporting more discomfort and greater difficulty achieving or maintaining an erection. Violent pornography consumption was associated with a greater likelihood of using sexual function enhancers. In women, watching pornography with physical violence was associated with a greater difficulty in reaching an orgasm in a relationship.

Regarding new STI diagnoses, 4 out of 10 men and 7 out of 10 women had been diagnosed with an STI in the past year. In both sexes, this diagnosis was more common among people who watch pornography with physical violence. The most commonly diagnosed STI among men was gonorrhea, followed by HPV. In women, the most common diagnosis was HPV, followed by chlamydia.

### 4.2. Violent Pornography Consumption

The results of this study have allowed us to delve into the violent content of pornography consumed by young adult women and men and identify how this content may be associated with affective-sexual competencies and sexual health. The results display a high number of men and women who watch pornography with physical violence. This high violent pornography consumption could reflect the high accessibility and presence of this content online ([26]). Regarding the frequency of violent pornography consumption, our results show higher percentages than those collected in previous studies ([21]). The different approaches to conceptualize violence in pornography ([13]; [51]), the measurement instruments ([64]) and the target population ([65]) could partly explain this higher rate of watching violent pornography in this study. The instrument used in our study intended to reduce desirability bias by avoiding words related to extreme physical violence, such as rape. The responses were collected using a Likert scale to encourage participants to describe the type of pornography they watched ([3]). Nonetheless, the target population in our study was a young adult population that used pornography and was sexually active. This population has been identified as one of the groups that consumes pornography the most ([54]).

### 4.3. Association of Violent Pornography Consumption with Assertiveness, Empathy and Acceptance of VAW

One of the objectives of this study was to analyze the association of exposure to different types of pornography with empathy, sexual assertiveness and acceptance of violence against women.

Our results partially confirm the hypothesis that men and women who view violent pornography have lower levels of assertiveness than those who view pornography with no explicit violence. These lower levels of assertiveness are identified in men exposed to pornography with physical violence but not in those exposed to degrading or humiliating pornography. Exposure to pornography with physical violence, the perceived perception of realism, can generate expectations far from reality in sexual encounters ([48]). This can negatively impact on the ability to effectively communicate sexual desires, boundaries and preferences, as well as the ability to negotiate consensual sexual practices and reject unwanted ones ([48]). This is an important result since assertiveness is a protective factor against victimization and sexual coercion ([23]; [49]) and risky sexual behaviours ([66]). Greater sexual assertiveness was associated with not only consistent condom use ([56]; [66]), but also the intention to use them ([63]), regardless of alcohol consumption ([75]). On the other hand, sexual assertiveness has also been positively associated with sexual satisfaction and sexual function ([53]).

Regarding women, our results showed that those who watched pornography with physical violence scored lower on assertiveness than those who watched pornography with no explicit violence, but this lower assertiveness was explained by the covariates included in the linear regression models. Although there was no association in our results between the type of pornography viewed and assertiveness in women, it is possible that assertiveness was a potential route for the victimization of female consumers regardless of the violent component of pornography ([19]; [20]; [24]; [62]).

Our results partially support the hypothesis that men, but not women, who view violent pornography have lower empathy than those who view pornography with no explicit violence. This correlation is identified in men who view pornography with physical violence but not in those who view degrading or humiliating pornography. It is possible that the association between empathy and online pornography consumption was bidirectional, and that, in turn, men with lower empathy consumed more pornography ([29]). Recent studies identify empathy as a mediating variable that may reduce -but not eliminate- the impact of online pornography on aggressive sexual behaviour ([55]) and partner violence ([29]). Violent pornography consumption and low empathy are included in the expanded version of the Confluence Model, which is widely used to identify factors that predict sexual aggression ([50]). In this model, the variables consumption of extreme pornography and low empathy emerge as predictors of aggressive sexual behaviour in men with high levels of hostile sexism ([50]). New discussions have emerged in recent years that identify empathy as a main skill in interpersonal relationships. [12] ([12]), in his three-step model of human empathy, proposed that empathy is not only related to the ability to put ourselves in someone else’s shoes, but also to the decision and justification for adopting the perspective of a particular person. This applies when being an observer or participant in situations involving multiple individuals. Under this model, pornography consumers empathize with one of the people in the scene and justify their position, leading to an empathetic disconnection from the other person. From this perspective, [6] ([6]) argue that men who frequently watch pornography automatically and emotionally identify with the actors because they experience the arousal and pleasure they seek in pornography. Additionally, they empathetically disconnect from women’s experience. The ability to decide with whom to empathize could explain women’s greater emotional resilience to the effects of violent pornography.

Our results confirm the hypothesis that men who view pornography with physical violence and/or degrading or humiliating pornography score higher in acceptance of violence against women than those who view pornography with no explicit violence. In the case of women, only those who view pornography with physical violence score higher in acceptance of violence against women than those who view pornography with no explicit violence. This possible greater effect of violent pornography on men’s attitudes could be because pornography is predominantly designed for male consumption, focusing on men’s pleasure and portraying women as objects of desire ([27]). Consumers’ perception of reality as a main variable in social learning ([7]; [48]), as well as individual values and attitudes associated with sexism and inequality between women and men, could intensify the internalization of violent sexual scripts towards women ([50]). Likewise, greater acceptance of violence against women among pornography users has been linked to a greater predisposition to sexual aggression and victimization among pornography users ([52]).

### 4.4. Association of Violent Pornography Consumption with Well-Being, Sexual Function and STIs

Research conducted to date on the possible association between the use of different types of pornography and sexual function is heterogeneous and does not allow us to formulate hypotheses on the direction of the association. Our results showed, in men, a positive association between watching pornography with physical violence and the likelihood of enjoying and/or having fun during sexual activities, but it was also found a positive association with the likelihood of experiencing greater discomfort and difficulties achieving or maintaining an erection. The prevalence of erectile dysfunction in men under 40 has increased in recent decades, rising from 2–5% in 1999 to 22% in 2021 ([59]; [42]). Studies conducted with sexually active young adults aged between 18 and 35 years have identified an association between the probability of having erectile dysfunction and problematic pornography use ([42]). Other studies analyzed the potential association between the frequency of pornography use and sexual function, without identifying a consistent one ([10]; [30]). In parallel, sexual dissatisfaction has recently increased. Some authors have associated this dissatisfaction with the use of pornography ([60]), although there were also studies that showed heterogeneous results associating pornography with greater sexual satisfaction ([38]). Some authors suggest that the content of pornography could partly explain this heterogeneity ([15]). Although there are studies that show that the variability of consumed content is very high and that content influences the level of arousal ([33], [34]), few studies have analyzed how violent content influences sexual function. To the best of our knowledge, there is currently only one study that analyzes how pornographic content is associated with sexual satisfaction and sexual function ([60]). For both men and women, the results showed that watching power and/or control pornography, as well as hardcore pornography, was negatively associated with sexual function in men, but not in women. Failure to fulfil sexual fantasies created by violent pornography could affect self-esteem and satisfaction, cause feelings of distress, and influence sexual function ([31]; [60]). These fantasies could cause sexual scripts in men that are not achievable in real life, potentially generating moral incongruence, and feelings of guilt and shame that interfere with sexual satisfaction and sexual function ([31]; [60]).

Furthermore, it is likely that men with sexual dysfunction watch more violent pornography in order to achieve a higher level of arousal. It is important to note the high consumption of sexual function enhancers observed in this study, particularly among men who consume violent pornography. In 50% of cases, these enhancers are taken without a medical prescription. Previous studies have indicated that recreational use of sexual enhancers in young adults is between 4.8% ([79]) and 22% ([8]). Although studies on this subject are scarce, research shows a concomitant consumption of alcohol and other drugs ([2]; [79]), and reduced condom use ([79]).

Concerning new STI diagnoses, our results partially support the hypothesis that exposure to violent pornography is associated with a higher prevalence of STIs in both sexes. Although these are descriptive data and should be interpreted with caution, a high number of recent diagnoses were predominantly reported by individuals who watched pornography with physical violence. Although the results of this study cannot be compared to the cases of nationally registered STIs, epidemiological surveillance systems in Spain report an increase in STIs mainly in men aged between 25 and 34 since the beginning of the 2000s ([37]). According to our results, epidemiological surveillance systems in Spain warn of a high incidence of STIs, with an increase in gonorrhea of 43% between 2021 and 2023 and 24.1% in the case of syphilis ([37]). Once the covariates were included in the logistic regression models, our study showed that men who watched pornography with physical violence had a higher probability of having been recently diagnosed with an STI, although this did not reach statistical significance (*p* = 0.084). The higher frequency of pornography consumption observed in men who viewed PPhV may partly explain these results. In line with our results, the studies by Kim identified an association between online pornography consumption and STIs in 16 to 18-year-olds ([44]). Studies by [82] ([82]) have shown that pornography consumption was associated with a lower frequency of condom use in both men and women who perceived pornography as a source of sexual information ([82]). In this sense, [36] ([36]) systematic review identified an association between pornography consumption, unsafe sexual practices, and a higher number of sexual partners.

### 4.5. Limitations

The results of this study must be understood considering their strengths and limitations. The cross-sectional nature of the study does not allow us to establish causal relationships or disregard the bidirectionality of the associations. Consequently, the directionality of the observed associations cannot be determined, nor can potential bidirectional relationships be ruled out (e.g., individuals with lower empathy may be more likely to consume violent pornography, or vice versa). Longitudinal or experimental studies are needed to establish causal inferences. The prevalence of men and women who report having watched violent pornography may be biased by desirability. Therefore, a measurement instrument was used to minimize this bias, and the regression models were adjusted with the social desirability scale. The instrument used to assess sexual well-being and function, while applicable to both women and men, has only been previously evaluated in women The items related to sexual function included in the male sample have not been previously evaluated; however, reliability values were acceptable and similar for both women and men. The measures used to assess empathy and sexual assertiveness are items from validated scales, not full scales, which may limit the interpretation of our results given the complexity of the constructs analyzed. Despite this, the reliability of the measures used in our study has Cronbach’s alpha values ranging from 0.59 to 0.94.

Since this study analyzes a subsample of the original study, specifically sexually active people who report having consumed pornography in the last 12 months, the statistical power of the analyzed sample may be less than initially estimated. This may make it difficult or underestimate the identification of possible associations. Finaly although several covariates were included (sex, age, education, sexual orientation, among others), other potentially relevant factors were not examined—such as insecure/ambivalent attachment patterns, internalized homophobia, sexism- which may influence the direction or strength of the associations found. Despite these limitations, the results obtained in this study are plausible and consistent with previous studies.

## 5. Conclusions

The findings of this study allow us to conclude that consumption of violent pornography might be associated with adverse outcomes across several dimensions of sexual health, encompassing both psychosocial competencies and physical and relational well-being. From a theoretical perspective, these findings support models emphasizing the role of pornographic content—particularly the Antecedent–Context–Effect framework—in shaping sexual attitudes and behaviors. They further reinforce the hypothesis that repeated exposure to violent sexual scripts may contribute to the normalization of coercion and gender inequality within intimate relationships.

### Future Research Directions

From a public health perspective, these results highlight the need to promote critical reflection on the messages, values, and attitudes transmitted by online pornography, as well as its consequences. Gender-sensitive educational interventions should strengthen empathy, communication, and sexual assertiveness while also addressing the regulation of violent pornography in the digital environment. These findings suggest that the consumption of violent pornography should be considered an emerging determinant of sexual and relational health among young adults.

## Figures and Tables

**Table 1 behavsci-15-01634-t001:** Sample description. Young adults (18–35 years old) who watch pornography and have had sexual relations in the last 12 months.

Variables	Man (*n* = 1249)		Woman (*n* = 736)		Total (*n* = 1985)	
	*n*	% *	*n*	% *	*n*	% *
University studies						
No	413	32.7	226	29.5	639	31.5
Yes	836	67.3	510	70.5	1346	68.5
Country of birth						
Spain	1209	96.8	637	86.9	1846	93.2
Abroad	40	3.2	99	13.1	139	6.8
Sexual orientation						
Heterosexual	967	77.7	594	81.2	1561	79.0
Bisexual	130	10.1	119	16.0	249	12.3
Gay/Lesbian	145	11.6	20	2.4	165	8.2
No answer	7	0.6	3	0.4	10	0.6
Partner in the last 12 months						
Stable and casual	150	12.1	66	8.7	216	10.8
Stable only	853	67.7	565	77.1	1418	71.2
Casual only	246	20.2	105	14.2	351	18.0
Type of pornography viewed in the last 12 months						
PNV	180	14.3	156	20.8	336	16.7
PDH	232	18.6	182	24.3	414	20.7
PPhV	799	63.7	368	50.7	1167	58.9
No answer	38	3.3	30	4.2	68	3.6
Pornography consumption frequency in the last 12 months						
Once a month or less	235	18.9	471	64.3	706	35.7
2–3 times a month	213	17.1	132	17.5	345	17.2
Once a week or more	779	62.1	110	14.9	889	44.7
No answer	22	1.8	23	3.4	45	2.4
	Mean *	SD *	Mean *	SD *	Mean *	SD *
Age (Years)	29.36	3.326	29	3.552	29.23	3.415
Social desirability	8.57	3.05	8.01	3.011	8.36	3.047
Total	1249	62.4	736	36.8	1985	100

Note. PNV: Exclusively pornography without explicit violence; PDH: Pornography with degrading/humiliating practices, no physical violence; PPhV: Pornography with physical violence; * weighted data; *n*: number; SD: standard deviation.

**Table 2 behavsci-15-01634-t002:** Mean item scores of the empathy, assertiveness and acceptance of violence scales.

Type of Pornography Consumed in the Past 12 Months (*n* = 1985; n.a_women = 30; n.a_men = 38)										
Dependent Variables	PNV (*n* = 336)			PDH (*n* = 414)			PPhV (*n* = 1167)			Effect Size
	*n*	Mean+	SD+	*n*	Mean+	SD+	*n*	Mean+	SD+	η^2^
Men (*n* = 1211)										
Empathy sum score	180	3.56	0.75	232	3.48	0.70	799	3.42	0.78	0.005
Assertiveness sum score (n.a = 3)	**179**	**4.43**	**0.66**	**230**	**4.44**	**0.61**	**799**	**4.24**	**0.77**	0.016
Accept-VAW sum score (n.a = 18)	**176**	**1.36**	**0.36**	**227**	**1.46**	**0.42**	**790**	**1.57**	**0.41**	0.037
Women (*n* = 706)										
Empathy sum score	156	3.89	0.61	182	3.83	0.62	368	3.85	0.69	0.001
Assertiveness sum score	**156**	**4.41**	**0.63**	**182**	**4.43**	**0.64**	**368**	**4.23**	**0.74**	0.018
Accept-VAW sum score	**155**	**1.31**	**0.31**	**182**	**1.31**	**0.34**	**368**	**1.44**	**0.40**	0.031

Note. PNV: Exclusively pornography without explicit violence; PDH: Pornography with degrading/humiliating practices, no physical violence; PPhV: Pornography with physical violence; + weighted data.; *n*: number; SD: Standard deviation; n.a: no answer; Numbers in bold indicate significant differences obtained using ANOVA tests.

**Table 3 behavsci-15-01634-t003:** Association between the consumption of different types of pornography and levels of empathy sexual assertiveness and acceptance violence in women and men: multivariate generalized linear regression models.

Type of Pornography Viewed (Reference: PNV)	Men				Women			
	Coefficient *	Lower Bound	Upper Bound	*p*	Coefficient *	Lower Bound	Upper Bound	*p*
	Empathy							
PDH	−0.029	0.013	−0.071	0.172	−0.011	−0.046	0.024	0.546
PPhV	−0.054	−0.015	−0.093	0.006	−0.002	−0.035	0.030	0.889
	Assertiveness							
PDH	0.005	0.033	−0.023	0.745	0.025	−0.006	0.055	0.113
PPhV	−0.034	−0.008	−0.061	0.012	−0.018	−0.048	0.012	0.236
	Acceptance of violence against women							
PDH	0.059	0.116	0.002	0.041	−0.003	−0.057	0.051	0.926
PPhV	0.122	0.170	0.073	0.000	0.076	0.029	0.122	0.002
	What rating would you give your regular sexual partner? Range from 0 to 10							
PDH	−0.002	0.040	−0.044	0.940	−0.010	−0.055	0.034	0.642
PPhV	−0.027	0.012	−0.065	0.170	−0.017	−0.053	0.019	0.367
	What rating would you give yourself as a regular sexual partner? Range from 0 to 10							
PDH	0.017	0.061	−0.028	0.460	0.051	−0.003	0.106	0.065
PPhV	0.011	0.049	−0.027	0.572	0.012	−0.038	0.061	0.640

Note. PDH: Pornography with degrading/humiliating practices, no physical violence; PPhV: Pornography with physical violence; * Coefficients adjusted by age, place of origin, sexual orientation, type of sexual partner, frequency of pornography consumption and social desirability.

**Table 4 behavsci-15-01634-t004:** Sexual health of women and men according to the violence in the pornography practices viewed.

	What Type of Pornography Do You View?					
	Men (*n* = 1249, n.a = 38)			Women (*n* = 736, na = 30)		
During Sexual Practices…	PNV (*n* = 180)	PDH (*n* = 232)	PPhV (*n* = 799)	PNV (*n* = 156)	PDH (*n* = 182)	PPhV (*n* = 368)
	% *	% *	% *	% *	% *	% *
…do you feel aroused?						
Not at all/A little/Somewhat	4.4	3.8	5.2	11.7	9.5	13.6
Quite a bit/A lot/Extremely	95.6	96.2	94.8	88.3	90.5	86.4
…do you have fun or enjoy yourself?						
Not at all/A little/Somewhat	7.2	4.7	3.7	9.1	10.0	10.1
Quite a bit/A lot/Extremely	92.8	95.3	96.3	90.9	90.0	89.9
…do you experience discomfort?						
Not at all/A little/Somewhat	**94.0**	**91.1**	**85.9**	76.0	73.3	68.5
Quite a bit/A lot/Extremely	**6.0**	**8.9**	**14.1**	24.0	26.7	31.5
…do you feel distress or fear?						
Not at all/A little/Somewhat	**91.7**	**91.9**	**86.3**	90.3	91.1	87.2
Quite a bit/A lot/Extremely	**8.3**	**8.1**	**13.7**	9.7	8.9	12.8
…generally, do you have orgasms?						
From Never to Sometimes	7.7	8.9	9.3	24.0	32.2	31.2
From Often to Always	92.3	91.1	90.7	76.0	67.8	68.8
Do you have difficulty reaching orgasm with a partner, but not alone?						
From Never to Sometimes	**95.0**	**96.2**	**89.4**	**81.2**	**76.0**	65.3
From Often to Always	**5.0**	**3.8**	**10.6**	**18.8**	**24.0**	34.7
If it were up to you, could you do without sexual practices?						
From Never to Sometimes	88.5	90.6	85.6	75.3	81.7	77.3
From Often to Always	11.5	9.4	14.4	24.7	18.3	22.7
Do you have difficulty getting or maintaining an erection?						
From Never to Sometimes	**92.8**	**93.2**	**85.3**			
From Often to Always	**7.2**	**6.8**	**14.7**			
Do you have premature ejaculation?						
From Never to Sometimes	93.4	95.7	91.3			
From Often to Always	6.6	4.3	8.7			
Do you use sexual function enhancers? (na = 3)						
No	98.4	93.2	90.3			
Yes	1.6	6.8	9.7			
Have you been diagnosed with any STI in the last 12 months? (na = 21)						
Yes	**1.1**	**2.1**	**5.7**	7.3	4.2	8.3
No	**98.9**	**97.9**	**94.3**	92.7	90.5	91.7
Total	14.9	19.2	66.0	22.1	25.8	52.1

Note. * weighted data; *n*: number; n.a: no answer; PNV: Exclusively pornography without explicit violence; PDH: Pornography with degradation/humiliation practices, no physical violence; PPhV: Pornography with physical violence. Numbers in bold indicate significant differences obtained using Chi^2^ tests.

**Table 5 behavsci-15-01634-t005:** Association between the consumption of violent pornography and sexual health in women and men.

Type of Pornography Viewed (Ref: PNV)	Men				Women			
	OR *	Sup	Inf	*p*	OR *	Sup	Inf	*p*
Do you feel aroused during sexual practices? (ref: Not at all/A little/Somewhat)								
PDH	0.95	0.30	2.95	0.925	1.28	0.62	2.66	0.510
PPhV	0.73	0.27	2.01	0.547	0.91	0.48	1.71	0.764
Do you have fun or enjoy yourself during sexual practices? (ref: Not at all to little/Somewhat)								
PDH	1.48	0.59	3.72	0.403	0.84	0.39	1.81	0.649
PPhV	2.47	1.13	5.42	0.024	0.85	0.42	1.74	0.661
Do you experience discomfort during sexual practices? (ref: Not at all/A little/Somewhat)								
PDH	1.37	0.59	3.17	0.46	1.08	0.65	1.81	0.768
PPhV	2.07	0.99	4.33	0.053	1.39	0.87	2.22	0.174
On occasion, do you feel distress or fear during sexual practices? (ref: Not at all/A little/Somewhat)								
PDH	1.06	0.47	2.43	0.885	0.79	0.37	1.72	0.56
PPhV	1.57	0.78	3.16	0.204	1.12	0.57	2.21	0.74
Generally, do you have orgasms during sexual practices? (ref: Never/Almost never/Sometimes)								
PDH	0.87	0.39	1.93	0.733	0.82	0.48	1.38	0.449
PPhV	1.00	0.50	2.00	0.992	0.90	0.55	1.47	0.680
Do you have difficulty reaching orgasm when you are with a partner, but not alone? (ref: Never/Almost never/Sometimes)								
PDH	0.69	0.26	1.84	0.458	1.15	0.66	2.00	0.62
PPhV	1.65	0.77	3.54	0.201	1.72	1.04	2.84	0.036
If it were up to you, could you do without sexual practices? (ref: Never/Almost never/Sometimes)								
PDH	0.87	0.43	1.75	0.688	0.66	0.38	1.13	0.125
PPhV	1.30	0.72	2.37	0.385	0.88	0.54	1.42	0.596
(Men) Do you have difficulty getting or maintaining an erection? (ref: Never/Almost never/Sometimes)								
PDH	0.87	0.39	1.93	0.732				
PPhV	1.85	1.00	3.43	0.052				
(Men) Do you have premature ejaculation? (ref: Never/Almost never/Sometimes)								
PDH	0.70	0.28	1.73	0.437				
PPhV	1.56	0.79	3.09	0.205				
(Men) Have you ever used any substance to enhance your sexual function? (ref: no)								
PDH	4.06	1.12	14.77	0.033				
PPhV	4.93	1.50	16.21	0.009				
Have you been diagnosed with any STI in the last 12 months? (ref: no)								
PDH	1.67	0.30	9.46	0.561	0.59	0.22	1.58	0.297
PPhV	3.79	0.83	17.25	0.084	1.01	0.47	2.18	0.971

Note. PDH: Pornography with degrading/humiliating practices, no physical violence; PPhV: Pornography with physical violence; * Adjusted OR by age, place of origin, sexual orientation, type of sexual partner, frequency of pornography consumption and social desirability.

## Data Availability

The data presented in this study are available upon request from the corresponding.

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
