# Peer review of "Violent Content in Online Pornography Is Associated with Sexual Health of Women and Men"

_behavsci, 2025, doi:10.3390/bs15121634_

Round 1

Reviewer 1 Report

Comments and Suggestions for Authors

Abstract and Presentation

  • Line 15: It is unnecessary to specify the types of pornography within the abstract; this level of detail appears too early. The abstract should succinctly present the study’s main variables, analyses, and key findings in a clear and engaging way. Excessive use of acronyms in the results section detracts from readability. Abstracts should attract readers’ interest by conveying the essence of the study in a simple yet scholarly manner, highlighting the dependent variables, main analyses, and outcomes relevant for readers deciding whether to continue reading.

Introduction and Formatting Issues

  • The introduction contains too many direct quotations at the beginning; paraphrasing would strengthen the academic tone.

  • Between lines 69 and 72, there is an inconsistency in font style, possibly due to copying and pasting.

  • Line 71 and Lines 89–90: The reference order appears incorrect and should be revised according to APA guidelines.

Methodology

  • The rationale for selecting the 18–35 age range is unclear; the authors should justify this choice.

  • A demographic questionnaire should be described to clarify which participant characteristics were collected.

  • The procedure for contacting approximately 9,000 participants also requires clarification: how were the email addresses obtained, and how did the researchers ensure participants were within the specified age range? Indicating whether a database, mailing list, or institutional resource was used would enhance transparency.

  • The description of participants and demographics would be more appropriately placed under the Participantssubsection of the Method section, ensuring methodological coherence.

Results

  • The results section is overcrowded with tables, many of which do not follow APA format. p-values should be listed beneath the tables, and statistically significant results clearly marked within them.

  • The type of regression analysis employed (“multivariate generalized regression models”) is ambiguous—authors should specify whether these were linear, logistic, or ordinal regressions.

  • Statements such as “association between watching P_PhV and lower empathy” should be rewritten for academic precision, e.g., “Men who watch P_PhV scored significantly lower on empathy measures.”

  • Statistical tests should be explicitly identified, and effect sizes reported.

  • Organizing the results by gender (e.g., Results for Men and Results for Women) would improve readability.

  • Some tables contain Spanish terms instead of English, which should be corrected. If tables are considered essential, they should be reformatted in APA style and possibly moved to an appendix to streamline the main text.

Discussion and Conclusion

  • The discussion spans nearly four pages but lacks a distinct conclusion. A concise conclusion section is necessary to highlight the study’s contributions and implications.

  • The discussion should focus on interpreting the findings rather than re-summarizing results or repeating the literature review. It would be more effective to begin by restating the study’s aims and then discuss the results in relation to relevant literature.

  • The limitations section appears incomplete; additional potential limitations and suggestions for future research should be added.

  • References are not formatted according to APA standards and should be revised for consistency.

Author Response

Please, see the attachment. Thank you very much. 

Reviewer 2 Report

Comments and Suggestions for Authors

The authors provide data on the association between pornography consumption—differentiating between violent and non-violent content—and several sexual health and psycho-relational parameters (such as empathy, acceptance of violence, etc.). The data are very interesting.
Overall, the paper is well written. However, it was not possible to view the questionnaire used to assess the type of pornographic material (the Supplementary Material file is not available). In the paper, I would suggest including an example item for each pornography category and asking the authors to specify what definition was provided to participants for each type of pornographic material. In other words: what is considered psychological violence? What is considered physical violence? If this clarification was not provided, the data collected may be considerably less “clean.”

I find the choice of using the terms “dependent variable” and “independent variable” somewhat debatable when referring to constructs that likely influence each other bidirectionally. It is plausible that watching violent pornography reduces empathy, but perhaps it is even more likely that individuals with low empathy are more inclined to watch violent pornography. I would therefore recommend using a more correlational approach.
That said, this does not detract from the overall interest and relevance of the data collected.

Below, I list a few additional minor suggestions aimed solely at improving the quality of the paper:

  • Lines 32–36: If this is a direct quotation, the page number should also be indicated.

  • Lines 37–41: I would suggest adding a citation.

  • Lines 69–76: The research data reported in these lines do not appear to be specifically related to the topic of sexual health. It seems less effective to include them right after a statement about studies concerning the association between pornography and sexual health. It might be more appropriate to move them to the previous paragraph on social/interpersonal skills, which in turn are connected to sexuality-related skills.

  • At the end of the introduction, the hypotheses could be stated more clearly, and the potential practical implications or usefulness of studying this topic could be elaborated further.

The authors chose to include participants aged 18–35. Why? I would suggest explaining the rationale for this choice.

If I understand correctly, the main analysis is based on the subsample of 1,985 individuals. Therefore, is the actual power of the study lower than that estimated in the original power analysis?

Regarding participant recruitment, it is unclear how the volunteers and the email addresses used for sending the invitations were obtained.

  • Line 120–121: “They were compensated with a points system that could be exchanged for items.” It is not clear what the authors mean here.

  • Lines 134–139: Where were these questions taken from? Were specific explanations or definitions provided for the terms used in the questions? For example, the following question seems rather unclear: “How would you rate yourself as a regular sexual partner on a scale of 1–10?”

  • Line 217: “Frequency of pornography use: once a week or more / 2–3 times a week / once a month or less.” This seems to be a typo. It probably should read “2–3 times a month.”

  • Line 234: I suggest adding the corresponding reference.

  • Line 290: What does “P_VF” stand for? Is this a typo?

In interpreting the data within the discussion, I would emphasize more strongly that the findings of the paper do not indicate causality, and that potential latent variables—such as insecure/ambivalent attachment patterns, internalized homophobia, etc.—could be intervening factors correlated with the “dependent variables” used in the study.

Round 2

Reviewer 1 Report

Comments and Suggestions for Authors

Thank you very much for carefully addressing the feedback and for revising the manuscript promptly and attentively. I appreciate the work you have put into improving the paper.

However, I still have several concerns that require clarification:

1. Measurement of the First Dependent Variable
You indicate that the measure used is the Instrument for Female Sexual Health and Dysfunctions in Primary Care. Yet, your sample includes male participants. This is confusing. Did you administer the same instrument to male participants as well? If so, this constitutes a methodological issue, as the scale was neither developed nor validated for male populations. Even though you mention adding several questions for men, the validation status of these added items is unclear. Adding items to an existing validated measure is generally not statistically appropriate, as it compromises the scale’s psychometric integrity.

2. Measurement of Cognitive and Affective Empathy
For the second variable, cognitive and affective empathy appear to be assessed using only a single item. Psychometrically, single-item measurement is not adequate for constructs of this complexity, and I do not think this approach is defensible from a statistical standpoint.

3. Psychometric Properties and Reliability
The statement that the measures “showed good psychometric properties” is not sufficiently academic or informative. Please report Cronbach’s alpha coefficients (or other reliability indices where appropriate) for each measure used in the study. This is essential, particularly given the adaptations and the inclusion of additional items.

4. Use of Abbreviations and Scale Descriptions
Before introducing any shortened forms (e.g., P_NV), please fully describe the construct and the scale. After that, use the shortened form consistently throughout the text. Also, in APA style, using “PNV” instead of “P_NV” is preferable for readability.

5. Presentation of Covariates and Measures
Instead of listing covariates in bullet-point form, it would be clearer and more academically appropriate to describe them in paragraphs, particularly the measurement details.

6. Tables and APA Style
All tables should conform to APA style guidelines. You may also consider shortening variable names in the tables (as you do in the text) to improve readability.
Additionally, several tables remain very long, which makes it difficult to follow the analyses alongside the results. While the final decision lies with the editor and other reviewers, some tables could be condensed or moved to supplementary materials.

7. Structure of the Discussion Section
Since the discussion already includes subheadings, you may consider adding subheadings for “Limitations” and “Future Research Directions” as well. This would improve clarity and navigation for readers.

Thank you again for your hard work and the considerable time you have invested in revising the manuscript. The paper has improved, but in my view, several points still require further clarification and methodological refinement.

Author Response

Thank you very much for your feedback again. Please see the attachment. 

Reviewer 2 Report

Comments and Suggestions for Authors

Congratulations to the authors.

The revised version of the manuscript is suitable for publication

Author Response

Thank you very much for your revision. 

Round 3

Reviewer 1 Report

Comments and Suggestions for Authors

Thank you to the authors for their thorough revisions and for addressing each of the points raised. A few of the references still need to be checked, as they do not fully adhere to the APA format. Beyond this minor issue, I have no further comments. Thank you.

Author Response

Dear revisor, 

Thank you very much for your review. The bibliography has been revised and it has been corrected. You can see the changes in the last manuscript version.